# Establishment of an Electro-Optical Mixing Design on a Photonic SOA-MZI Using a Differential Modulation Arrangement

**DOI:** 10.3390/s23094380

**Published:** 2023-04-28

**Authors:** Hassan Termos, Ali Mansour, Majid Ebrahim-Zadeh

**Affiliations:** 1ICFO—The Institute of Photonic Sciences, 08860 Castelldefels, Spain; majid.ebrahim@icfo.eu; 2Lab-STICC, CNRS UMR 6285, ENSTA Bretagne, 2 Rue François Verny, CEDEX 09, 29806 Brest, France; 3Institucio Catalana de Recerca i Estudis Avancats (ICREA), Passeig Lluis Companys 23, 08010 Barcelona, Spain

**Keywords:** electro-optical SOA–MZI system, differential modulation mode, up mixing function, quadrature amplitude modulation

## Abstract

We design and evaluate two experimental systems for a single and simultaneous electro-optical semiconductor optical amplifier Mach-Zehnder interferometer (SOA-MZI) mixing system based on the differential modulation mode. These systems and the optimization of their optical and electrical performance largely depend on characteristics of an optical pulse source (OPS), operating at a frequency of f= 39 GHz and a pulse width of 1 ps. The passive power stability of the electro-optical mixing output over one hour is better than 0.3% RMS (root mean square), which is excellent. Additionally, we generate up to 22 dBm of the total average output power with an optical conversion gain of 32 dB, while achieving a record output optical signal to noise ratio (*OSNR*) up to 77 dB. On the other hand, at the SOA–MZI output, the 128 quadratic amplitude modulation (128-QAM) signal at an intermediate frequency (IF), $f_{1}$, is up-mixed to higher output frequencies $nf\;\pm \;f_{1}$. The advantages of the resulting 128-QAM mixed signal during electrical conversion gains (ECGs) and error vector magnitudes (EVMs) are also evaluated. The performed empirical SOA-MZI mixing can operate up to 118.5 GHz in its high-frequency range. The positive and almost constant conversion gains are achieved. Indeed, the obtained conversion gain values are very close across the entire range of output frequencies. The largest frequency range achieved during experimental work is 118.5 GHz, where the EVM achieves 6% at a symbol rate of 10 GSymb/s. Moreover, the peak data rate of the 128-QAM up mixed signal can reach 70 GBit/s. Finally, the study of the simultaneous electro-optical mixing system is accepted with unmatched performance improvement.

## 1. Introduction

Radio over fiber (RoF) and millimeter-wave (MW) systems are of great importance, as they formally proffer a high optical transmission rate, light weight, low cost, and wide frequency range up to 2 THz with implementation in wireless fidelity (WiFi) communications, radar applications, and cellular networks [1,2,3,4,5]. Optical transmission systems can benefit from various features of optical mixing, including positive conversion efficiency at high output frequencies and low error vector magnitude (EVM) values with high data rates based on the nonlinearities of several optical mixers. Photonic mixers can also perform essential logic gate operations in addition to mixing [6,7,8,9,10,11,12,13,14,15,16]. These optical mixers include Mach–Zehnder modulators (MZMs) with good performance characterizations [5,17], electro-absorption modulators (EAMs) with the cross-absorption modulation (XAM) effect [18,19] used for mixing, Photodiodes (PDs), particularly uni-traveling carrier PDs (UTC-PDs) [20,21], and positive-intrinsic-negative PD (PIN-PD) [22], which are used for frequency mixing due to their nonlinear current response, semiconductor optical amplifier (SOA) reliant on the four-wave mixing (FWM) [23,24] and the cross gain modulation (XGM) [25,26] impacts, and SOA-Mach Zehnder interferometer (SOA-MZI) dependent on the cross-phase modulation (XPM) [27,28,29,30,31] effect.

SOA–MZIs are a superb candidate for up and down frequency mixing [32,33,34,35,36,37,38,39,40] due to their exceptional performance with single, cascaded, and parallel SOA-MZIs linkages, including their high frequency conversion range, high data transmission rate, and lofty conversion gain in both empirical and simulated considerations. The used SOA-MZI can be used for high-performance all-optical mixing or electro-optical mixing [41,42,43,44] by combining it with an optical pulse source (OPS) [45,46,47]. In fact, the OPS employs a mode-locked laser to create an optical pulse train with a small pulse width duration and a high repetition rate or sampling frequency. Furthermore, SOA-MZIs can be employed by all-optical sampling for simultaneous up and down frequency shifting [48].

A SOA can be used to complete electro-optical frequency mixing [41,42]. Recently, an effective explanation of an electro-optical mixing theory at high frequencies reliant on the reflective SOA (RSOA) is provided in [41]. In accordance with this theory, a low frequency signal carrying QPSK data has been moved to a high frequency of 15 GHz, utilizing a high performance system.

A SOA-MZI differential modulation mode-based electro-optical up-converter is presented in our recent work [44]. The SOA-MZI optical port is used to introduce the OPS signal to each SOA in the upper and lower arms, and its electrical port is used to introduce the intermediate frequency (IF) signal. Using a simulation program called Virtual Photonics Inc. (VPI) [49], we show that the suggested electro-optical mixing system has good characteristics, including a frequency range up to 195.5 GHz, a data rate of 5 GBit/s, and a very high conversion efficiency.

In this disquisition, we fervently provide a novel electro-optical mixing system based on a SOA-MZI differential modulation that was created by employing our earlier circuit [44]. Under this theory, the locations of the IF and OPS signals are exchanged. The replica quality of the up-mixed signal, which follows the OPS signal’s harmonics, can be optimized by placing the OPS signal at each SOA electrical gate. This outstanding improvement results from the characteristics of the OPS signal, which include a high repetition rate of 39 GHz and a brief pulse width duration of 1 ps. The OPS harmonics experience minor degradation because of these features. As a result, the reproductions of the sampled signals exhibit little regression with respect to the output frequency at the SOA-MZI output. Therefore, the novel experimental electro-optical mixing system, which is evaluated for the first time in the optical domain and the electrical field, has outstanding qualities, specifically power stability, a high optical signal to noise ratio (*OSNR*), a frequency mixing range up to 118.5 GHz for the single mixing and 120 GHz for the simultaneous mixing, approaching substantially higher conversion gains (CGs), and the demodulation of the 128-QAM (Quadrature Amplitude Modulation) up-mixed signal at the bit rate of 70 GBit/s through error vector magnitude (EVM).

## 2. SOA-MZI Module

In our experimental platform, we used a SOA-MZI produced by the center for integrated photonics (CIP) company (40G-2R2-ORP) [45,50]. The optical module, which comprises a nonlinear SOA in each arm of the interferometer, is used by the CIP instruments as an integrated hybrid apparatus to experimentally detect the electro-optical mixing regime. This SOA-MZI can be used to perform wavelength conversion and optical and electro-optical logic gate operators. The SOA-MZI also offers certain significant features, including the ability to operate at high data rates, C-Band compatibility, low insertion loss, and high output powers. Four input optical and two input electro-optical connectors and two output optical connectors are also included in the SOA-MZI. Using complementary metal oxide semiconductor (CMOS) manufacturing, the SOA-MZI has been developed to produce useful photonic integrated circuits (PICs) [51,52,53,54].

A Peltier cooler and a thermistor are included in the SOA-MZI module for essential temperature control. Electrical pins are included in the applied SOA-MZI for each phase shifter and the bias current of each SOA. To properly enable phase control of the MZI, autonomous thermo-optic phase shifters are electrically isolated and combined. The resistance of a complete phase shifter is approximately 100 ohms (Ω). Since the phase shifters are resistive heaters, they need to be powered by a voltage supply source. The SOA-MZI operational temperature must remain constant throughout all measurements in order for stationary capabilities to be met.

## 3. The Principle of the Updated Electro-Optical Mixing

Figure 1 illustrates how a SOA-MZI differential modulation principle evolved into an electro-optical mixing system. This updated concept is developed using information derived from the pragmatic study discussed in [44]. The input signals are exchanged between the electrical and optical input ports with various operating point properties, which is the only distinction between them.

At the electrical port of each SOA, the control OPS (optical pulse source) signal is inserted at a repetition rate of f. Using a photo diode (PD), this optical signal is converted to an electrical one, which is then amplified by a low noise amplifier (LNA), before being injected at the electrical gates of the used SOA-MZI. Direct current (DC) bias and the control OPS signal are used to control each SOA electrode. In contrast, an optical coupler (OC) transmits the data signal at frequency $f_{1}$ across the SOA-MZI’s upper and lower arms. According to this theory, the OPS signals can use the bias current to adjust each SOA carrier density and refractive index. Additionally, due to the direct dependence of the SOAs optical gain on their carrier density, the electrical OPS signal modulating the bias current in both arms magnificently induces a nonlinear gain amendment. Subsequently, the OPS signal modulates the intermediate frequency (IF) signal in the uppermost and lowermost arms. Due to the SOA-MZI nonlinearities, which include cross gain modulation (XGM) and cross phase modulation (XPM) consequences, the modulated optical gains with the amplified optical output powers are successfully obtained at the outturn of each SOA in order to achieve the optical mixed signal.

Compared to the antecedent notion [44], the up-mixed signal at the SOA-MZI output is produced with outstanding quality [44]. The up-mixed signal, obtained by an optical spectrum analyzer (ESA) in the electrical domain, has replicas at the output frequencies of $nf\pm f_{1}$, induced by the square-law revelation [25,55] following photo-detection by a PD and LNA amplification. The advantage of this electro-optical regime is that the harmonic power level of the control OPS signal is kept constant. As a result, the power of the mixed signal at any given frequency is conserved at the same replica level. The up mixed signal at the intended frequencies $nf\pm f_{1}$ will, therefore, roughly have the same overall efficiency (similar electrical conversion gains) and performance quality (as the error vector magnitudes (EVMs)). It is important to mention that, to the best of our knowledge, this is the first time the current approach has been experimentally proven.

## 4. Performance of the Electro-Optical Mixing System for an Up-Mixed Signal

A.Experimental Setup Characterization

Figure 2 depicts the experimental configuration of the current electro-optical up-mixer built on the real SOA-MZI differential modulation regime. In this system, a mode-locked fiber laser produced by Pritel Inc. is used as the optical pulse source (OPS) [45,46,47]. At the electoral port of each SOA, an OPS signal with a wavelength of λ= 1550 nm and a repetition rate of f= 39 GHz is employed, in addition to the OPS structure. Before being delivered to the SOA’s electrical port, this signal with a pulse width of 1 ps, is captured by a PhotoDiode (PD) and then amplified by a low noise amplifier (LNA). The first three harmonics of the OPS signal have about the same amplitude in the electrical domain and are present at the frequency $H_{n}=nf$, as shown in Figure 3. Because we have the same OPS parameter characteristics, which optimize the operation of the electro-optical mixing system, the spectrum at the electrical port of each SOA becomes similar. The third harmonic $H_{3}$ and the first harmonic $H_{1}$ amplitudes only differ by 0.5 dB. Additionally, both SOA electrical ports are set to 60 Ω, and the bias currents of SOA1 and SOA2 are $I_{1}=I_{2}=$ 400 mA. It is noteworthy to note that, when SOAs are biased at 400 mA, the utilized SOAs have an active zone distance of 750 mm and the highest optical gain of 28 dB.

On the other hand, the optical data signal, which is generated by a laser diode device at a wavelength of $\lambda_{1}=$ 1540 nm, is intensity-modulated by an optical Mach-Zehnder modulator (MZM) under the control of an electrical 128-QAM (Quadrature Amplitude Modulation) subcarrier at a frequency of $f_{1}=$ 1.5 GHz. Additionally, the MZM is biased in its linear region and has a 6 GHz electrical field bandwidth [45]. To reduce the nonlinearity effects where electro-optical mixing is performed, the MZM must operate with a very low modulation index.

An arbitrary waveform generator (AWG) at the electrical gate of the optical Mach Zehnder modulator (MZM) generates 128-QAM at a variety of symbol rates.
Figure 4
shows the electrical spectrum of the data signal at the AWG output with a symbol rate of 1 GSamp/s. Additionally, as shown in
Figure 5, the average optical power of the data signal is $P_{IF}=-$ 10 dBm, which, at 1.5 GHz, is equivalent to the electrical power of $P_{f_{1},i}=-$ 31.5 dBm after photo-detection by the PD and, subsequently, electrical amplification by the LNA.

At the outturn of the SOA–MZI, the optical band pass filter (OBPF) is regulated at 1550 nm to get rid of undesirable signals. The focal wavelength of the applied OBPF designed at the SOA-MZI outturn is calibrated at 1550 nm to handpick the optical up mixed signal forged by the used interferometer while eliminating the optical data signal centered at 1540 nm. Thus, in order to produce an up-mixed signal with exceptional performance due to the XGM nonlinearity, the data signal is converted to the wavelength of the control OPS signal. Additionally, the applied OBPF’s bandwidth is predetermined to be 0.65 nm in order to greatly reduce the amplified spontaneous emission (ASE) noise produced by both SOAs while approving the OPS harmonics. This approach involves up-converting the data signal so that the OPS harmonics are followed by replicas of the up-mixed signal at output frequencies up to 118.5 GHz. The optical loss caused by the OBPF is 3.5 dB, as well. At the SOA-MZI exit, the filtered up mixed signal is precisely detected in an optical spectrum analyzer (OSA) before being converted to the electrical up mixed signal, which is the required signal and should be monitored in this study.

Afterwards, the optical up-mixed signal is applied through a PD for the electrical transmogrification. The experimental setup uses an InGaAs/InP positive-intrinsic-negative PDs (PIN-PDs) epitaxial heterostructure to convert optical signals to electrical signals with a fixed responsivity. PIN-PDs are compatible with semiconductor devices that are quick to react to both photons and high-speed particles. According to the incident optical power, they function by assimilating photons or charged particles to produce a current flow in an extrinsic network [56,57]. All photo diodes in our experimental setup have a sensitivity of 0.85 A/W and a bandwidth of 300 GHz. In order to achieve the desired spectrum of the up mixed signal, the photo-detected signal is then amplified by a 33 dB-gain LNA and then shown on an electrical spectrum analyzer (ESA). It is vital to note that the LNA’s primary function is to increase the electrical power level of the received signal to a sufficient level above the noise floor so that it can be used for additional mixed processing. Furthermore, the 128-QAM up mixed signal is digitalized using the digital sampling oscilloscope (DSO). Then, the 128-QAM up mixed signal is demodulated using a vector signal analyzer (VSA) software program in order to determine its EVM (error vector magnitude) values.

B.Optical Characterizations

(a)Output Power Performance

The output optical power of the up-mixed signal contingent on the electro-optical differential modulation system using a sampling SOA-MZI mixer is measured: $P_{RF}=$ 22 dBm.

(b)Optical Conversion Gain

Inequality between the output mixed power $P_{RF}$ and the input data power of $P_{IF}=-$ 10 dBm, as shown in Equation (1), serves as a demonstration of the optical conversion gain (*OCG*) for up-mixing characterizations. Thus, the *OCG* value of 32 dB is a great measurement by which to assess the electro-optical differential modulation system.
(1)$$OCG=P_{RF}-P_{IF}$$ where
$P_{IF}$
is the input optical power of the intermediate frequency (IF) signal at the SOA-MZI input and
$\;P_{RF}$
is the output optical power of the radio frequency (RF) signal that is the up mixed signal at the SOA-MZI output.

(c)Power Stability

The empirical outcomes of the long-term power stability metrology of the up mixed signal for the modernized electro-optical mixing system at the SOA-MZI yield are displayed in Figure 6. The output power is measured to indicate outstanding passive stability greater than 0.3% RMS (root mean square) for up-mixing over a period of nearly an hour. The optical up mixed signal at a focal wavelength of 1550 nm at the SOA-MZI output is achieved to determine the enduring power stability.

(d)Signal to Noise Ratio

An important consideration when assessing any optical system is the optical signal to noise ratio (*OSNR*) [58]. Numerous systems, including wireless ones and radio over fiber (RoF), use the *OSNR*. It is elucidated as a variation in dB between signal and noise powers $\left(P_{IF\;}\&\;P_{N_{i}}\;\right),$ identified in dBm at the SOA-MZI input and between signal and noise powers $\left(P_{RF\;}\&\;P_{N_{o}}\right)$ at the SOA-MZI output, as described in Equations (2) and (3), respectively.
(2)$$OSNR_{i}=P_{IF}-P_{N_{i}}$$
(3)$$OSNR_{o}=P_{RF}-P_{N_{o}}$$ where $OSNR_{i}$ is the optical signal to noise ratio of the data signal at the SOA-MZI input, $OSNR_{o}$ is the optical signal to noise ratio of the up mixed signal at the SOA-MZI output, $P_{N_{i}}$ is the input noise power of the data signal at the SOA-MZI input, and $P_{N_{o}}$ is the output noise power of the up mixed signal at the SOA-MZI output.

Additionally, the OSNRs are monitored for the data signal and the up mixed signal at both the SOA-MZI input and the SOA-MZI exit, respectively. As previously indicated, the optical power at the SOA-MZI output and the SOA-MZI input are $P_{RF}=$ 22 dBm and $P_{IF}-$ 10 dBm, respectively, while the noise power at the SOA-MZI output and input are $P_{N_{o}}=-$ 55 dBm and $P_{N_{i}}=$
− 92 dBm, respectively. Because of this, the input OSNR
$\left(OSNR_{i}\right)$ and the output one $\left(OSNR_{o}\right)$ are 82 and 77 dB, correspondingly. Based on the electro-optical mixing system using the SOA-MZI differential modulation, the *OSNR* peak value of the up mixed signal of 77 dB is considered excellent. It is significant to note that the optical up mixed signal is impacted by amplified spontaneous emission (ASE) noise from both SOAs, which introduces noise into the up mixed signal throughout the amplification and mixing processes. Numerous types of noise, including shot noise, ASE noise, and thermal noise, are produced when an optical signal is converted by a PD into an electrical signal [59].

(e)Noise Figure

The noise figure $\left(NF\right)$ [60,61] is another excellent parameter for quantifying the electro-optical mixing system, dependent on the SOA-MZI differential modulation. It is interpreted as a variation in dB of the input OSNR
$\left(OSNR_{i}\right)$ at the SOA-MZI input to the output OSNR
$\left(OSNR_{o}\right)$ at the SOA-MZI exit, as identified in Equation (4). The operating wavelength, the bias current, and the input optical signal power of the input data signal all affect how well the SOA-MZI mixing system performs in terms of noise figure $\left(NF\right)$ [62,63,64]. The resulting NF, which is computed from the *OSNR* at the input and output of the SOA-MZI described earlier, is 5 dB, which confirms the electro-optical mixing system’s successful operation.
(4)$$NF=OSNR_{i}-OSNR_{o}$$

C.Electrical Characterizations

(a)Electrical spectrum

We provide the electrical spectrum of the up-mixed signal at the SOA-MZI outer gate, as shown in Figure 7. It can be shown that the data signal at $f_{1}=$ 1.5 GHz is up-mixed at output frequencies $nf\pm f_{1}$.

Additionally, there is a slight retrogression of the OPS signal’s harmonics. Less than 1 dB separates the first harmonic at f= 39 GHz from the last harmonic at 3f= 117 GHz. That is why the replicas of the up-mixed signal have roughly the same electrical power.
The OPS characteristics, including the repetition rate of 39 GHz and the pulse width duration of 1 ps, as well as the modulation of the data signal by the OPS signal in both arms, are attributed to these novel experimental findings. Finally, the optical band pass filter (OBPF), which is set at the OPS wavelength of 1550 nm at the SOA-MZI output, also plays a role in evaluating the system.

(b)Electrical Conversion Gain

The electrical conversion gain (*ECG*) is understood as the difference of electrical powers measured in dBm between the up mixed signal at $nf+f_{1}$ on the right of harmonics at the SOA-MZI outturn and the input data signal at $f_{1}$ at the SOA-MZI inlet. *ECG* allows the valuation of contemporary electro-optical differential modulation systems, as illustrated in
Equation (5).
(5)$$ECG=P_{nf+f_{1}}-P_{f_{1},i}$$ where $P_{f_{1},i}$ is the electrical power of the input data signal at a frequency $f_{1}$ at the SOA-MZI inlet, and $P_{nf+f_{1}}$ are the electrical powers of the up-mixed signal at output frequencies $nf+f_{1}$ at the SOA-MZI exit.

The upgraded electro-optical mixing system has a great improvement in the ECGs, which are essentially constant across the increase in the output frequency up to $3f+f_{1}=$ 118.5 GHz, see Figure 8. The highest frequency range of the electro-optical mixing system based on the SOA-MZI differential modulation corresponds to the output frequency of $3f+f_{1}=$ 118.5 GHz, where the *ECG* reaches 38.7 dB. At $f+f_{1}=$ 40.5 GHz, this value is 40 dB. Therefore, the difference between them is only 0.3 dB. This demonstrates how the innovative SOA-MZI differential modulation improves the electro-optical mixing system’s efficacy at higher output frequencies. The enhancement of the harmonics of the OPS control signal and SOAs gain is what causes the replicas of the up-mixed signal to improve.
(6)$$ECG=P_{nf+f_{1}}-P_{f_{1},i}$$ where $P_{f_{1},i}$ is the electrical power of the input data signal at a frequency $f_{1}$ at the SOA-MZI inlet, and $P_{nf+f_{1}}$ are the electrical powers of the up-mixed signal at output frequencies $nf+f_{1}$ at the SOA-MZI exit.

(c)Isolation

At the SOA-MZI output, isolation measured in dB is shown to be an equivalence of electrical powers obtained in dBm between the OPS signal that is frequently used as a local oscillator (LO) signal and the up-mixed signal at a radio frequency (RF), which is called LO-RF isolation $\left(I_{LO-RF}\right)$, or it is between the input data signal at an intermediate frequency (IF) and the up mixed signal at a radio frequency (RF), which is known as IF-RF isolation $\left(I_{IF-RF}\right)$, as displayed in Equations (7) and (8), respectively.
(7)$$I_{LO-RF}=P_{nf}-P_{nf\pm f_{1}}$$
(8)$$I_{IF-RF}=P_{f_{1},o}-P_{nf\pm f_{1}}$$
where $P_{nf}$ is the electrical powers of the OPS signal at frequencies nf. $P_{f_{1},o}$ is the electrical power of the input data signal at a frequency $f_{1}$. $P_{nf\pm f_{1}}$ are the electrical powers of the up mixed signal at frequencies $nf\pm f_{1}$. All the mentioned powers are considered at the SOA-MZI output.

In Figure 9, LO-RF and IF-RF isolations are detected. The relationship between the up-mixed signal at the output frequencies $nf+f_{1}$ and the OPS signal at frequencies $H_{n}=nf$ is known as the LO-RF isolation, which decreases slightly with the output frequency. The IF-RF isolation, which increases somewhat from 1.5 dB at 40.5 GHz to 1.8 dB at 118.5 GHz, is also a relationship between the input data signal at $f_{1}$ and the up mixed signal at $nf+f_{1}$. In other words, despite certain degradations in the OPS signal’s harmonic strengths and replicas of the up mixed signal at the SOA-MZI outrun, both isolations are essentially stable.

(d)Error Vector Magnitude

The up-mixed signal loading 128-QAM data is measured at a variety of symbol rates, ranging from 1 to 10 GSymb/s for the first and third replicas at the lowest and highest frequencies of 40.5 and 118.5 GHz, respectively. An arbitrary waveform generator (AWG) at the optical MZM’s electrical port creates the 128-QAM data, which is then used as an input of the SOA-MZI. The error vector magnitude (EVM) [58,65,66], measured by a vector signal analyzer (VSA) application after digitalizing by a digital sampling oscilloscope (DSO), is used to evaluate the effectiveness of the electro-optical mixing system. Additionally, the EVM values are viewed as a gauge of 128-QAM modulation precision [67].

The EVM of the 128-QAM up mixed signal expands with the symbol rate for the first replica at $f+f_{1}=$ 40.5 GHz, as well as for the third one at $3f+f_{1}=$ 118.5 GHz, as bestowed in Figure 10. At 118.5 GHz, the EVM value hits 6%, while, at 40.5 GHz at 10 GSymb/s, it is 5%. Additionally, the 128-QAM up-mixed signal’s top bit rate rises to 70 GBit/s. Thanks to the cutting-edge electro-optical differential modulation technology, the important task of improving the up-mixed signal is played by tracking the harmonics of the optical pulse source (OPS) signal.

Figure 11 shows the constellation pattern of the 128-QAM up mixed signal. At the greatest output frequency of $3f+f_{1}=$ 118.5 GHz at 70 GBit/s, the EVM value is 6%. This diagram is an example of a constellation diagram that the vector signal analyzer (VSA) software has successfully produced.

## 5. The Performance Parameters of the Electro-Optical Mixing System for Simultaneous Two up Mixed Signals

Figure 12 shows the experimental setup of the simultaneous two up-mixed signals well established on the electro-optical mixing system, employing SOA-MZI differential modulation. Currently, it involves the second data signal (channel two), which is introduced concurrently with the first data signal (channel one), utilizing an optical coupler (OC) to the SOA-MZI’s optical middle port. The characteristics of channel two are the average optical power of $P_{IF2}=-$15 dBm, the wavelength of $\lambda_{2}=$ 1545 nm, the intermediate frequency of $f_{2}=$ 3 GHz, and the electrical power of $P_{f_{2},i}-$ 40 dBm. In order to achieve the concurrent mixed signals at the SOA-MZI output, which is the primary objective of this revised experimental setup compared to the prior one, the OPS signals primarily modulate the simultaneous data signals in the SOA-MZI arms. The performance characterizations of simultaneous two-up-mixed signals in the optical and electrical domains will also be examined.

The optical filter, which is adjusted at 1550 nm for the categorization of the optical properties of the simultaneous electro-optical mixing system, transforms the simultaneous two data signals at the SOA-MZI output to the OPS signal’s wavelength. The output power of the simultaneous mixing is, therefore, 24 dB, which is 2 dB more than the output power of the single mixing. Additionally, there are two optical conversion gains (*OCG*s) associated with channels one and two. These *OCG*s are 34 and 39 dB, respectively. The optical to noise ratios (OSNRs) for channels one and two at the SOA-MZI input, which were appropriately measured at central wavelengths of 1540 and 1545 nm, are 82 and 87 dB, respectively, while the optical concurrent up-mixed signal at the SOA-MZI exit is 79 dB when considered at the central wavelength of 1550 nm. As a result, two noise figure (NF) values are obtained, one of which is clearly defined as the difference between the output *OSNR* and the input *OSNR* connected to channel one, and the other is the difference between the output *OSNR* and the input *OSNR* connected to channel two. They are 3 dB and 9 dB, respectively.

The electrical spectrum of the simultaneous up-mixed signals for the electro-optical mixing system, as shown in Figure 13, clearly verifies the contemporaneous electro-optical mixing principle and shows four replicas around each harmonic of the OPS signal. As a result, the two combined input data signals are converted into concurrent up-mixed signals at output frequencies $nf\pm f_{k}$, where k is an integer with only two possible values connected to channels one and two. Because channel two has a low average optical power at the SOA-MZI inlet despite having a higher intermediate frequency, the replicas connected to channels one and two have extremely similar amplitudes. On either side of the harmonics of the OPS signals, there is only a 0.5 dB difference between the replicas related to channels one and two.

Additionally, as shown in Figure 14, the SOA-MZI frequency mixing achieves good results with low input power needs due to the extinction ratio (ER) of the optical peak transmitted power, which is higher by 2.4 dB at −15 dBm compared to −10 dBm at the SOA-MZI output. The second channel, on the other hand, has a greater intermediate frequency, which results in lower efficiency. That is a compromise between the operational points of channels one and two. As a result, the replicas connected to channel one exhibit experimental findings that are similar to those connected to channel two.

The electrical conversion gain $\left(ECG\right)$ of the simultaneous up-mixed signals depends on the electro-optical mixing system and is presented in Figure 15. It is measured for the third replica at output frequencies $3f+f_{k}$ allied to channels one and two. The ECGs of the third replica at $3f+f_{1}$, which is connected to channel 1, are similar, as propounded in Figure 8, with a 0.5 dB deterioration across the entire output frequency range. Furthermore, the ECGs of the third replica at $3f+f_{2}$ linked to channel two are higher by 9.5 dB compared to the ones related to channel one. This improvement is the result of the characteristics of channel two at the SOA-MZI input, particularly its electrical power, which is 10 dB lower than that of the first channel. On the other hand, at the SOA-MZI output, both channels are up-shifted, with almost the same electrical power. This is due to the SOA-MZI being subjected to a mixing optimization based on electro-optical differential modulation at −15 dB. Furthermore, as previously shown in Figure 14, the second channel has an ER of 26.2 dB, which is higher than the extinction ratio (ER) of channel one, which is 23.8 dB.

Figure 16 displays the EVM measurements of the two 128-QAM up-mixed signals operating simultaneously. Additionally, it is derived for the third replicas using output frequencies $3f+f_{k}$ and various symbol rates. The contiguous electrical powers of the two up-mixed signals in the electrical domain are 0.5 dB greater for the up mixed signal connected to channel one. Because of this, over the whole symbol rate range, the up-mixed signal at $3f+f_{2}$ converted from channel two has an error vector magnitude (EVM) that is 1% larger than the up-mixed signal at $3f+f_{1}$ transformed from channel one. The achievement quality of channel two is also excellent, especially in terms of its input *OSNR*. When channel one and channel two are combined, the mixing process, which is dependent on the electro-optical SOA-MZI differential regime, is accomplished with closer optical and electrical fineness at the SOA-MZI output. The top bit rate of the simultaneous two 128-QAM up mixed signals likewise increases to 70 GBit/s.

## 6. Conclusions

In this article, a novel SOA-MZI differential modulation-based empirical electro-optical mixing scheme is discussed. The optical and electrical domains are used to investigate this system. Several metrics, including optical conversion gain (*OCG*), optical signal to noise ratio (*OSNR*), noise figure (NF), electrical conversion gain (*ECG*), and error vector magnitude (EVM) were used to quantify the findings of the experiments. Optical characterizations show that the resulting output up-mixed signal has excellent passive power stability, with a RMS value of 0.3% over an hour. Additionally, we were able to attain an *OSNR* of 77 dB at the SOA-MZI output, which is good in an electro-optical arrangement, as well as an *OCG* of up to 32 dB. In electrical characterizations, the *ECG* and EVM values have mostly been used to assess the effectiveness, as well as the quality, of the electro-optical mixing system. Furthermore, a wide range of output frequencies is reached for high positive ECGs. At 10 GSymb/s, the 128-QAM electro-optical mixing displays exceptional EVM values up to 6%. This electro-optical mixing study has a maximum frequency range of 118.5 GHz and a maximum bit rate of 70 GBit/s. The performance of the electro-optical SOA-MZI mixer is optimized by improving the output power of the replicas, which is made possible by improving the characteristics of the OPS signal, including the high repetition rate and short pulse width duration with high SOAs gains. Finally, the electro-optical SOA-MZI differential modulation-based simultaneous mixing is validated with improved performance. This powerful new electro-optical mixing technology can, therefore, be used with a wide range of modernistic implementations.

## Figures and Tables

**Figure 1 sensors-23-04380-f001:**
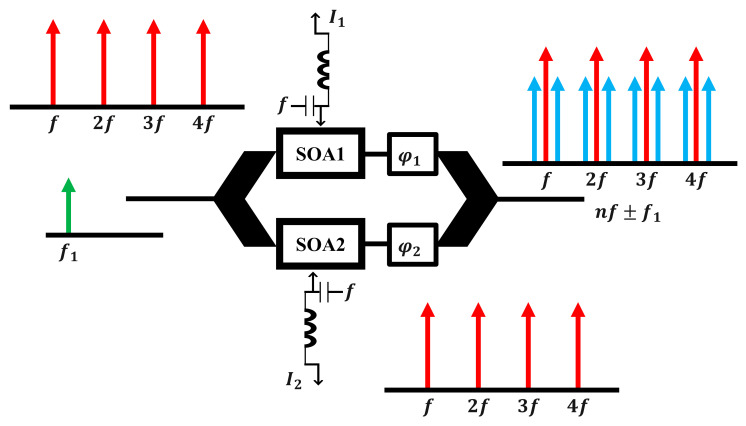
Electro-optical mixing schematic derived from a SOA–MZI differential modulation regime. $I_{1}$ is the SOA1 current, $I_{2}$ is the SOA2 current, $\varphi_{1}$ is the upper phase shifter, $\varphi_{2}$ is the lower phase shifter, n is an integer, f is the repetition rate of the OPS signal, $f_{1}$ is the intermediate frequency of the data signal, and SOA: semiconductor optical amplifier.

**Figure 2 sensors-23-04380-f002:**
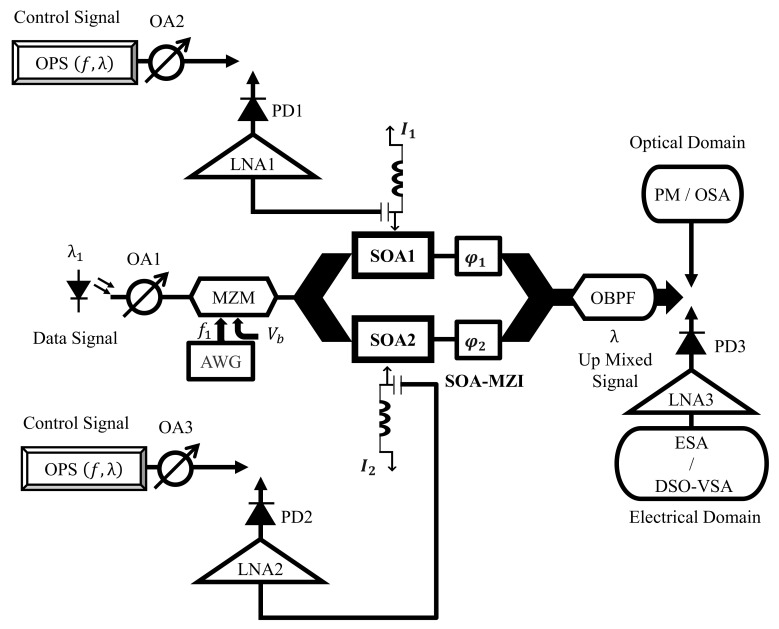
Experimental setup of the novel electro-optical mixing dependent on a genuine SOA–MZI differential modulation regime. ESA: electrical spectrum analyzer, OSA: optical spectrum analyzer, OA: optical attenuator, OBPF: optical band pass filter, AWG: arbitrary waveform generator, LNA: low-noise amplifier, PD: photo diode, PM: power meter, DSO: digital sampling oscilloscope, VSA: vector signal analyzer, SOA-MZI: SOA-Mach Zehnder interferometer, $V_{b}$: bias voltage, and OPS: optical pulse source.

**Figure 3 sensors-23-04380-f003:**
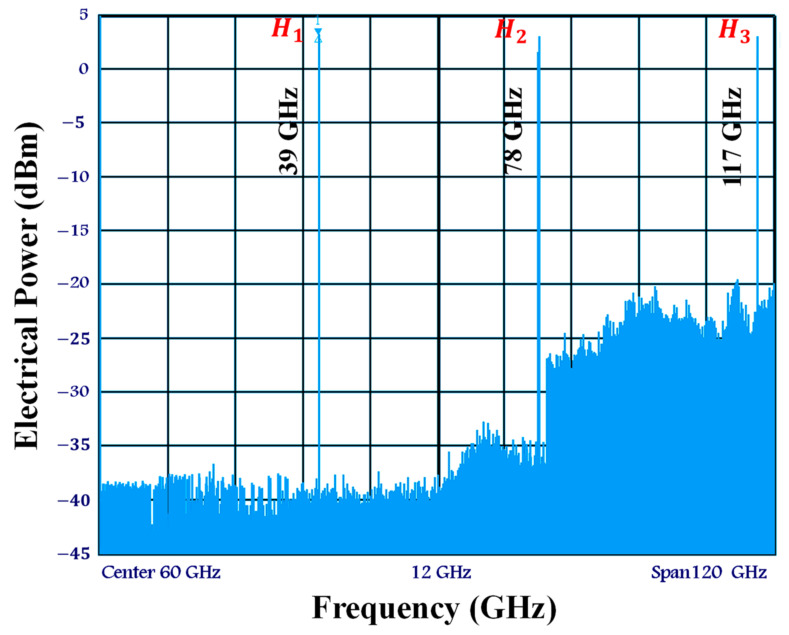
Electrical spectrum of the control OPS signal at the electrical port of each SOA at frequencies $H_{n}=nf$.

**Figure 4 sensors-23-04380-f004:**
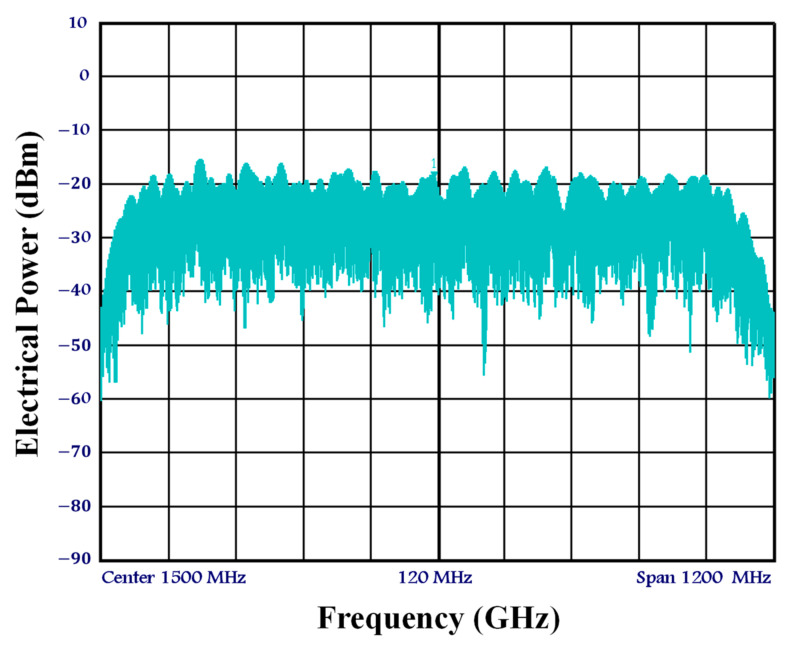
Electrical spectrum of the data signal at the arbitrary waveform generator (AWG) output at a symbol rate of 1 GSymb/s.

**Figure 5 sensors-23-04380-f005:**
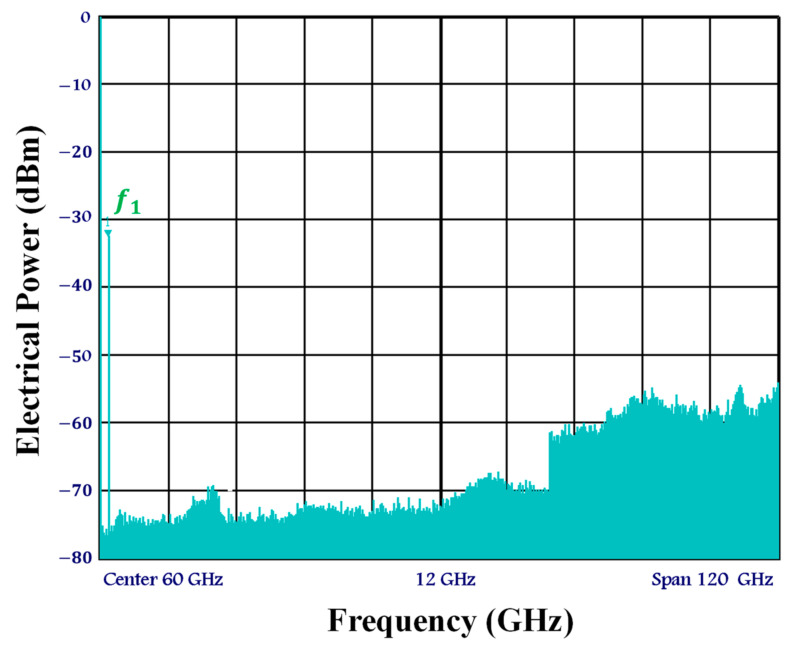
Electrical spectrum of the data signal at the SOA-MZI middle port at a frequency of $f_{1}$.

**Figure 6 sensors-23-04380-f006:**
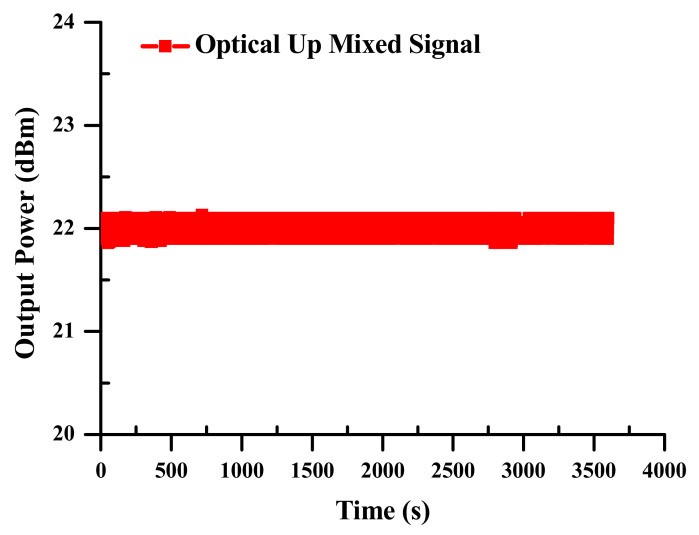
Power stability of the up-mixed signal at the SOA-MZI output.

**Figure 7 sensors-23-04380-f007:**
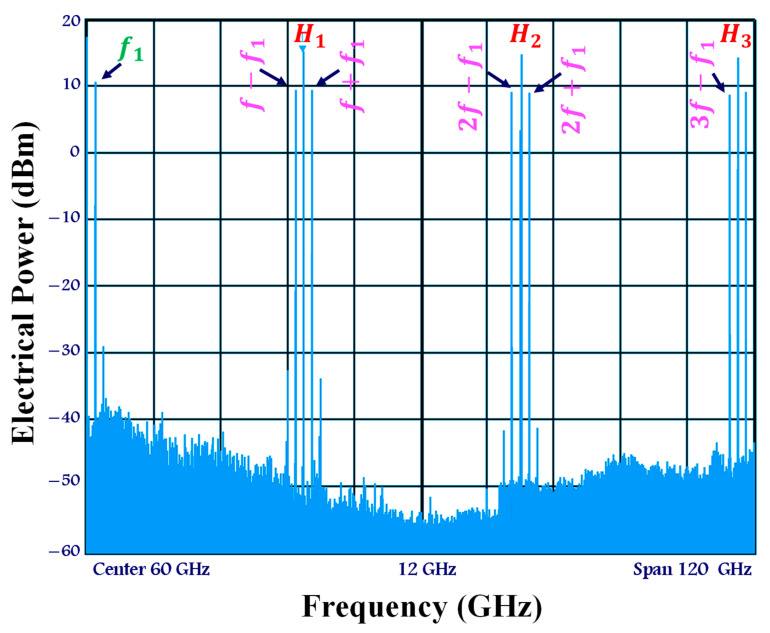
Up-mixed signal spectrum at the SOA-MZI output at output frequencies $nf\pm f_{1}$ originated from the input data signal at $f_{1}$.

**Figure 8 sensors-23-04380-f008:**
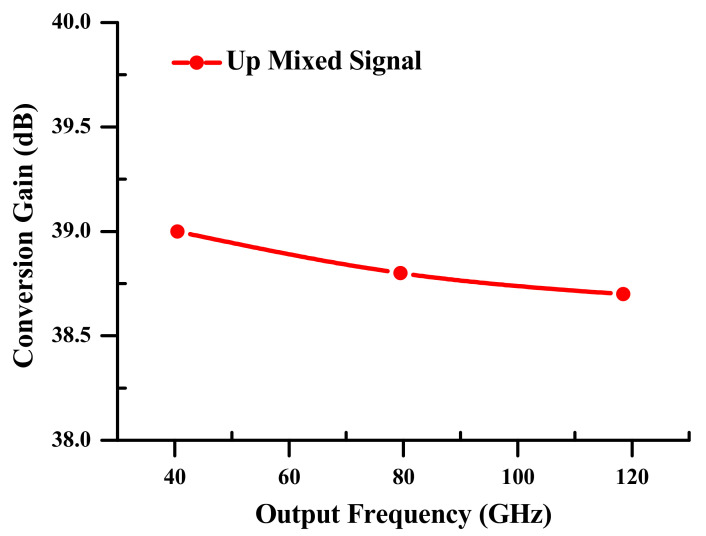
Electrical conversion gains (ECGs) of the electro-optical mixing system contingent on the differential SOA-MZI modulation mode at $nf+f_{1}$.

**Figure 9 sensors-23-04380-f009:**
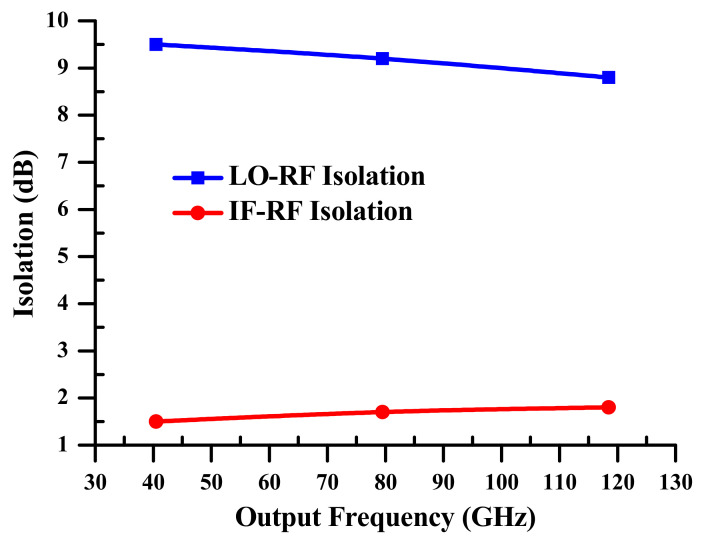
LO-RF and IF-RF isolations at the SOA-MZI output. LO: local oscillator, IF: intermediate frequency, and RF: radio frequency.

**Figure 10 sensors-23-04380-f010:**
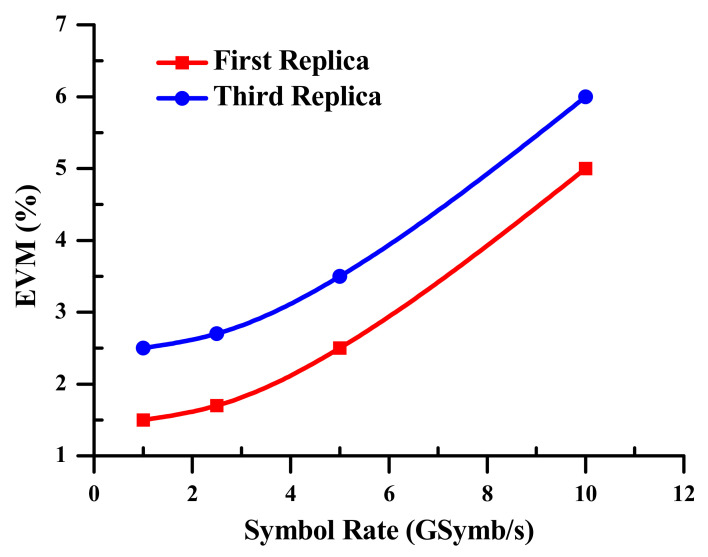
EVM measurements of the up-mixed signal for the first replica at $f+f_{1}$ and the third replica at $3f+f_{1}$.

**Figure 11 sensors-23-04380-f011:**
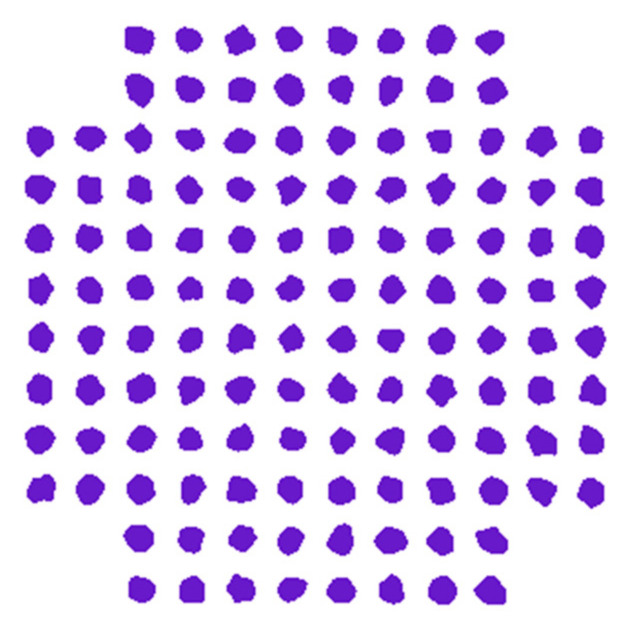
Constellation diagram of the 128-QAM up-mixed signal at the outbound port of the applied SOA-MZI.

**Figure 12 sensors-23-04380-f012:**
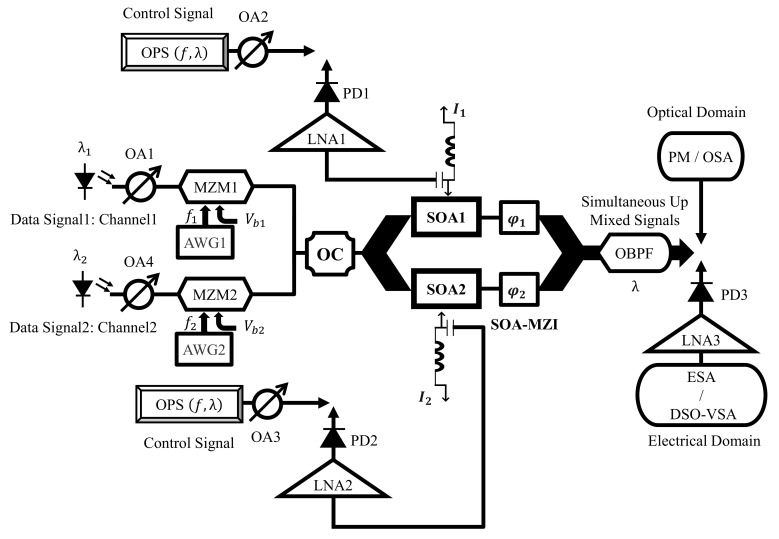
Experimental setup of the simultaneous electro-optical mixing dependent on a SOA–MZI differential modulation regime for two concomitant up-mixed signals. The only difference in comparison to the previous setup in Figure 2 is that two data signals are injected together at the optical input gate of the SOA-MZI in order to obtain simultaneous mixing. OC: optical coupler.

**Figure 13 sensors-23-04380-f013:**
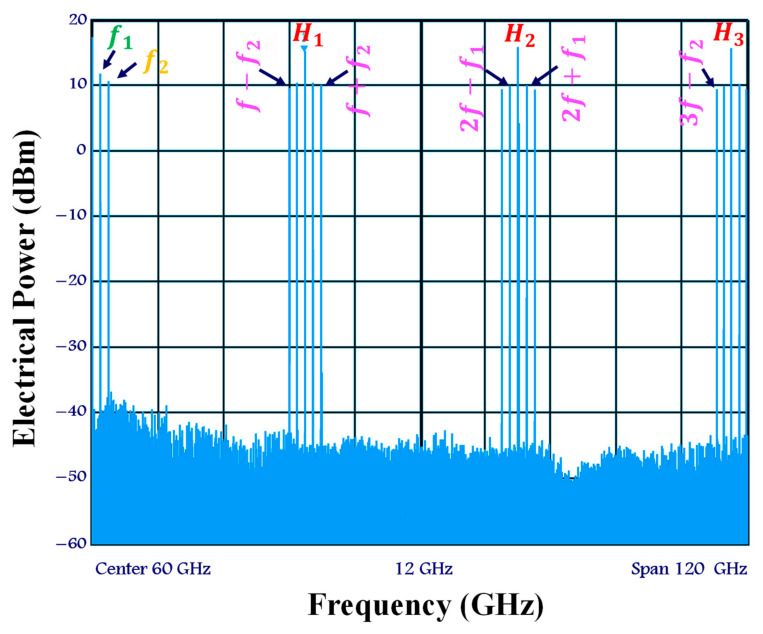
The electrical spectrum of simultaneous up-mixed signals, depending on the electro-optical mixing regime.

**Figure 14 sensors-23-04380-f014:**
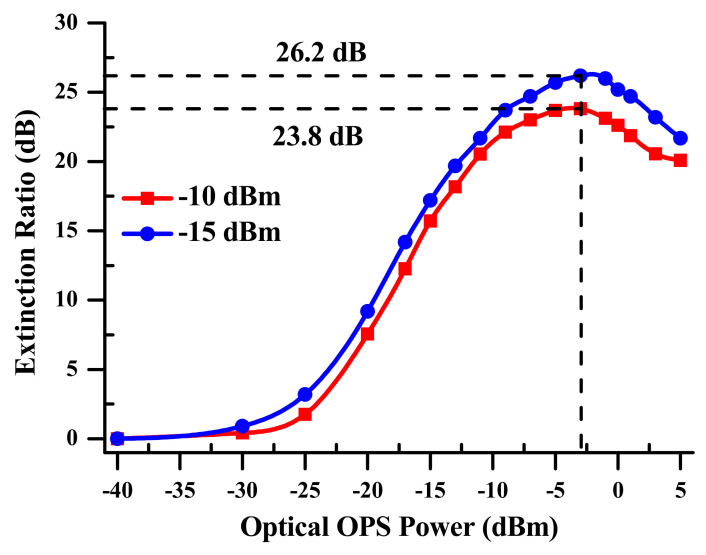
SOA-MZI static behaviors at two different average optical powers of −10 dBm and − 15 dBm related to channel one and channel two, respectively.

**Figure 15 sensors-23-04380-f015:**
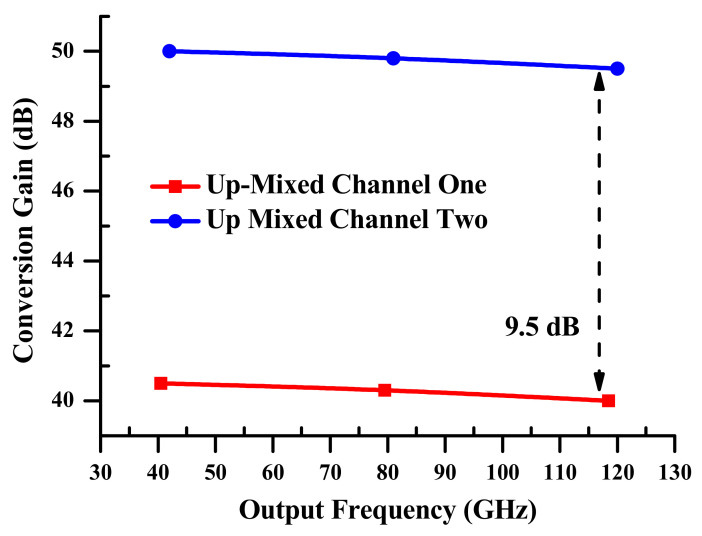
Electrical conversion gains of the simultaneous up mixed signals at the SOA-MZI output as a function of the output frequency, which is measured for the third replica at $3f+f_{k}$.

**Figure 16 sensors-23-04380-f016:**
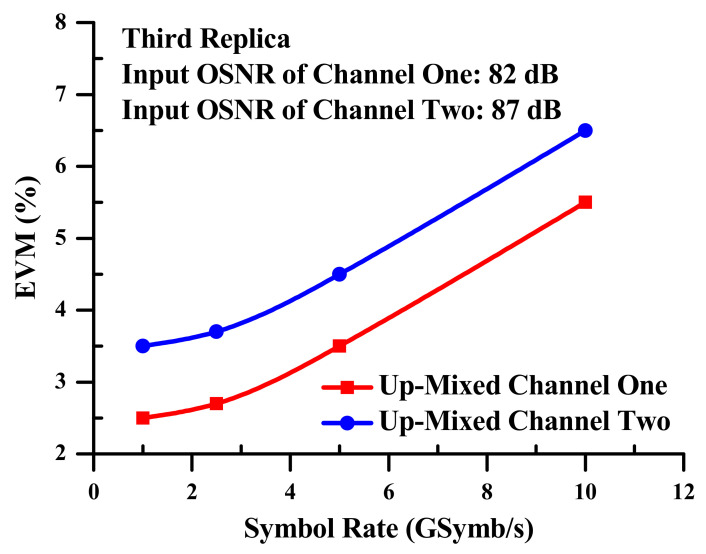
EVM measurements of the simultaneous 128-QAM up mixed signals.

## Data Availability

Data underlying the results presented in this paper are not publicly available at this time, but they may be obtained from the authors upon reasonable request.

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
