# Peer review of "Establishment of an Electro-Optical Mixing Design on a Photonic SOA-MZI Using a Differential Modulation Arrangement"

_sensors, 2023, doi:10.3390/s23094380_

Round 1

Reviewer 1 Report

The authors propose a new approach to the design of SOA-MZI mixers. The OPS is now converted into driving signals to the SOAs using photodiodes and low noise amplifiers, while the data signal is sent to the common port of the MZI. The new arrangement allows for improved performance, in particular high signal to noise ratio (77dB) for the mixed signals, high conversion gain and good power stability. The authors also demonstrate simultaneous mixing of 2 signals at different frequencies.

The manuscript proposes an original approach to SOA/MZI mixing. The work is well presented and proposes interesting discussions. Publication is recommended. A few comments below should be addressed:

Can the authors elaborate more on the device fabrication? Can they provide a picture of the integrated photonics device? What material is used? What is integrated on the device (LNA, PDs, OPBF, OA)? What is external?

Were the authors able to measure the input/output128-QAM constellations? If so a plot here would be useful.

Proof-reading + editing is recommended before publication.

Author Response

Dear Professor,

Thank you for your comments.

Reviewer 2 Report

In this paper, an Electro-Optical Mixing System is proposed. Based on this, the experiment was carried out. In the experiment, the 128-QAM IF signal was mixed up to 118.5GHz and the data result of the experiment is analyzed. However, there are some problems that need to be solved.

1.     The explanation of "the Principle of the updated Electro-Optical Mixing" is not detailed enough. For example, how does the control OPS apply to SOA? the up mixed signal at purpose frequencies ?? ± ?1 how come?

2.     What is the specific role of OBPF in the experiment? Can it be illustrated in the optical spectrum?

3.     Are the notes“Up mixed Channel Two” in Figure 12 formatted incorrectly?

4.     Please add the electrical spectrum or optical spectrum of the 128-QAM signal in the experiment.

5.     In Figure 12 and Figure 13, the colors of “Up-Mixed Channel One” are suggested be the same.

Author Response

(The authors gave the same response as above.)

Reviewer 3 Report

The manuscript deals with design and characterization of electro-optical mixing device for high frequency application. From the scientific point of view the work may be eligible for publishing, even though it is an extension of previous works, but the presentation of the results is not clear enough.

As a first, there is an extensive usage of abbreviations which disrupt a fluent reading of the content. For example the abbreviation IF (intermediate frequency) is used only few times in the whole document and it takes some time to find explanation when it firstly appears. I do not see any sense of using this abbreviation and it is even used with mistake in Figure 1 caption, where "... the IF frequency ..." is written. The abbreviation AWG for example has several meanings in electronic and photonic field of research (Arbitrary Waveform Generator, Arrayed Waveguide Grating, American Wire Gauge), and its usage may be therefore very confusing. Some of the abbreviations make sense, such as SOA-MZI, but some does not. I would recommend to review the usage of abbreviations in the manuscript to reduce it to acceptable number.

The English level of the content is bellow average, and hard to read. Some translations are questionable, such as "...becomes acclimatized to 60 Ω." on page 3. More correct wording would be "...is optimized for 60 Ω." or "is matched for 60 Ω.". Or "... is inaugurated by an arbitrary..." on page 4 could be rather "...is generated by an arbitrary ...". There are many more, but I will not introduce all of them here. For example, two sentences on page 5 does not give any sense to me "Furthermore, the LNA will have unstable mixing process over more the period of 64 minutes and as a result it oscillates at some frequencies. Furthermore, the LNA will have unstable mixing process over more the pe-riod of 64 minutes and as a result it oscillates at some frequencies. When the mixed signal carries 128-QAM data, we capitalize on the digital sampling oscilloscope (DSO) and vector signal analyzer (VSA) in order to demodulate the 128-QAM up mixed signal and obtain its EVM (Error Vector Magnitude) values.". It seems like google translator output without revision. Even the title of the manuscript is not very clear to me how it is related to the manuscript content. There is an extensive number of wrong written words. I recommend to revise the English grammar.

The realization of the experiment is very interesting and the results may be considered as impressive, but the presentation of the results degrades the whole experiment and is not worth of publishing in Journal with actual impact factor of 3.8. The optical measurements are not trivial and there is certain information about the electro-optical device and measurements missing. For example how were the optical signals coupled to the photonic circuit? What is the real appearance of the electro-optical mixing device and which materials is it made of? Better description of used components is missing. In present form it is a black-box which has been used for experiment. In author's PhD thesis (cited 41) is device and workplace from 2017, but is it the same? I would appreciate a picture of actual workplace and device directly in the manuscript, because going through 183 pages of the thesis to find the information is not very joyful. For me it lacks scientific discussion about the obtained results, but it seems to be trend in recent publications. For discussed reasons, I do not recommend to publish the paper in present form and will consider it after revision.

Author Response

(The authors gave the same response as above.)

Round 2

Reviewer 3 Report

The manuscript has been significantly improved in the revised version. Thank you for the answers about the measurements, it shall be in the paper for the reader to better understand the results. I appreciate the list of key abbreviations at the beginning of the manuscript  and improvement of the English grammar. Despite of some residual mistakes in text I am recommending the manuscript to be accepted for publication.